# AKD Emulsions Stabilized by Guar Gel: A Highly Efficient Agent to Improve the Hydrophobicity of Cellulose Paper

**DOI:** 10.3390/polym15244669

**Published:** 2023-12-11

**Authors:** Xiaona Liu, Yingpu Li, Huili Wang, Zhaoping Song, Congping Tan, Guodong Li, Dehai Yu, Wenxia Liu

**Affiliations:** 1Key Laboratory of Pulp and Paper Science & Technology of Ministry of Education/Shandong Province, State Key Laboratory of Biobased Material and Green Papermaking, Faculty of Light Industry, Qilu University of Technology, Shandong Academy of Sciences, Jinan 250353, China; liuxn0107@126.com (X.L.); liyingpu1@163.com (Y.L.); pingzi221@163.com (Z.S.); tcp7411@163.com (C.T.); lgd73758557@163.com (G.L.); yudehai@qlu.edu.cn (D.Y.); liuwenxia@qlu.edu.cn (W.L.); 2Bloomage Biotechnology Corporation Limited, Jinan 250013, China

**Keywords:** AKD emulsions, guar gel, stability, packaging paper, hydrophobicity

## Abstract

The aim of the present study was to investigate highly efficient alkyl ketene dimer (AKD) emulsions to improve the hydrophobicity of cellulose paper. AKD emulsions stabilized by guar gel were obtained; the guar gel was prepared by hydrogen bond cross-linking sodium tetraborate and guar gum. The cross-linking was confirmed by combining FTIR and SEM. The effect of guar gel on the performance of the AKD emulsions was also studied by testing AKD emulsions stabilized by different guar gel concentrations. The results showed that with increasing guar gel concentration, the stability of the AKD emulsions improved, the droplet diameter decreased, and the hydrophobicity and water resistance of the sized packaging paper were gradually enhanced. Through SEM, the guar gel film covering the AKD emulsion droplet surface and the three-dimensional structure in the aqueous dispersion phase were assessed. This study constructed a scientific and efficient preparation method for AKD emulsions and provided a new method for the application of carbohydrate polymer gels which may avoid the adverse effect of surfactant on paper sizing and environmental problems caused by surfactant bioaccumulation.

## 1. Introduction

The hydrophobicity of cellulose paper is an important characteristic that is usually realized by the addition of hydrophobes in the papermaking process, and it is conceptually difficult to add these hydrophobes to paper. It is widely popular to employ the sizing process to treat fibers with hydrophobic additives to decrease the rate of aqueous liquid penetration into paper and paperboard. Alkyl ketene dimer (AKD) and alkenyl succinic anhydrides (ASA) are two popular internal sizing agents, which by definition produce paper with hydrophobic properties. AKD is a waxy, water-insoluble solid, and the reaction group of AKD is the lactone ring formed by the dimerization of ketene. It is common understanding that the AKD lactone ring adsorbed on the fiber reacts with the hydroxyl group on the fiber forming the bond of β-keone ester which makes AKD stick to the fibers. The hydrophobic end of the AKD molecule is toward the outside of the fiber, thus increasing the hydrophobicity of the fiber and improving the water resistance of the paper (sizing degree). Because AKD is a solid oil at room temperature with a melting point between 40 and 60 °C, it should be emulsified before its addition into the fiber’s slurry [1]. AKD is a solid dictating that any emulsion be formed above its melting temperature in the presence of surfactants. After cooling to room temperature, liquid AKD droplets become solid particles that engage in a solid–water interfacial interaction for sizing. Surfactants have played a decisive role in the emulsification process for a long time; however, particle stabilizers have recently attracted a great deal of attention in the field of sizing emulsions because of their excellent interface stability, nontoxicity, lack of enrichment of anionic impurities, and lack of foaming in the flow system [2]. Rigid colloids and soft gel particles have both been used as particle stabilizers and have drawn more attention. Particle stabilizers absorbed at the oil–water interface can form a dense particle film, reducing interfacial tension and increasing the emulsion stability. In recent years, researchers have been investigating the mechanism of emulsions stabilized by particles, and their potential applications in many industries, such as cosmetics, food and pharmaceuticals [3]. A variety of particles, including, chitosan [4], gelatin [5], layered double hydroxide [6], laponite [7], cellulose nanofibrils [8], and bentonite [9], have been successfully utilized for preparing sizing emulsions in papermaking. At present, the emulsifiers of AKD mainly include cationic starch [10], synthetic cationic polymer emulsifier with certain surface activity [11], and styrene propylene polymer [12]. When emulsifying using cationic starch and cationic polymer, a lot of surfactants are usually required. Otherwise, high energy emulsification technology such as high-pressure homogenization must be employed to acquire the appropriate particle size emulsion, and the amount of cationic starch or synthetic cationic polymer is very high, usually 1–4 times that of AKD. Using cationic starch or synthetic polymer aqueous solution with high viscosity will inevitably increase the viscosity of AKD emulsion, and therefore, it is hard to prepare high AKD content emulsion, which would also cause both high energy input and low production efficiency [6,7]. Furthermore, the presence of surfactant in the emulsion could reduce the efficiency of AKD sizing and increase the frothing of the slurry which is harmful to paper uniformity [6,7]. Non-surfactant AKD emulsion is expected and a broader range of practice applications is desirable, which can be attributed to a high sizing efficiency and high storage stability.

Guar gum is a natural nonionic galactomannan extracted from the endosperm of the guar bean [13,14]. Carbohydrate polymers, guar gum, and cellulose fibers readily form hydrogen bonds through hydroxyl groups, and thus guar gum can be used as a wet-end chemical additive to improve the physical and mechanical properties of paper [15]. In addition, guar gum can react with borate, metal ions, and other cross-linking agents to generate a viscoelastic material or membrane [16,17]. Borax cross-linked guar gel has three-dimensional network structures, a multi-stimulus response, and high self-healing properties, which can be used in many fields [16,17,18]. It has been reported that guar gum can be combined with surfactant or polymer [19] to prepare colloidal systems as emulsifier-stabilized emulsions [20], but there are few reports on the preparation of emulsions with borax cross-linking guar gum. Theoretically, the good film-forming property of guar gel is conducive to the coverage of the emulsion droplets, and the three-dimensional network structure is beneficial for improving the stability of the emulsion. However, to the best of our knowledge, guar gel used to stabilize AKD emulsions has not been reported to date.

In this study, we aim to investigate highly efficient AKD emulsions to improve the hydrophobicity of cellulose paper. AKD emulsions stabilized by guar gel were obtained. Guar gel was used as the emulsifier of the AKD emulsions and was prepared by mixing borax and a guar gum aqueous solution at room temperature. The cross-linking was confirmed by combining FTIR (Fourier transform infrared spectroscopy) and SEM (scanning electron microscopy). The effect of guar gel on the performance of the AKD emulsions was also studied by testing AKD emulsions stabilized by different guar gel concentrations. The results showed that with increasing guar gel concentration, the stability of the AKD emulsions improved, the droplet diameter decreased, and the hydrophobicity and water resistance of the sized packaging paper were gradually enhanced. The guar gel film covering the AKD emulsion droplet surface and the three-dimensional structure in the aqueous dispersion phase were assessed using SEM. This study constructs a scientific and efficient preparation method for AKD emulsions and provides a new method for the application of carbohydrate polymer gels which may avoid the adverse effect of surfactant on paper sizing and environmental problems caused by surfactant bioaccumulation.

## 2. Materials and Methods

### 2.1. Materials

Guar gum (GG, Mw 352.76) was purchased from Shanghai Macklin Biochemical Technology Co., Ltd. (Shanghai, China). Sodium tetraborate decahydrate (borax) was purchased from Tianjin Kermel Chemical Reagent Co., Ltd. (Tianjin, China). AKD powder was provided by Suzhou Tianma Specialty Chemicals Co., Ltd. (Suzhou, China). Aspen alkaline peroxide mechanical pulp (APMP) was purchased from Dezhou ZMSY Pulping Co., Ltd. (Dezhou, China). All chemicals were used without further purification. Deionized water was used in the emulsion process.

### 2.2. Preparation and Characterization of Guar Gel

A 0.5 wt% guar gum solution was obtained by dissolving guar gum in deionized water and stirring at 40 °C for 30 min. The guar gel was prepared by mixing 1.000 g borax and 100 mL guar gum solution, stirring for 30 min, and sealing for 2 h at room temperature.

The freeze-dried guar gel was ground in an agate mortar, pressed with KBr into discs, and then measured using a Fourier transform infrared spectrometer (FTIR, Nexus670, Thermo Fisher Scientific Inc., Waltham, MA, USA) in the range of 400–4000 cm^−1^.

In addition, the size and morphology of the freeze-dried guar gel were investigated using scanning electron microscopy (SEM, HITACHI Regulus8220, Tokyo, Japan).

### 2.3. Preparation of the AKD Emulsions

To prepare the AKD emulsions, solid AKD and guar gel aqueous dispersion (the ratios of guar gel to AKD were 0.25, 0.5, 0.75, 1.0, and 1.25%, respectively.) were first heated in a 75 °C water bath. With an AKD to water mass ratio of 1:7, the oil and water phases were mixed and emulsified using an emulsifier (YULDOR Y25 High Shear Emulsifier, Guangzhou, China) at 10,000 rpm for 3 min. A series of fine AKD emulsions were obtained and designated as AKD emulsion—0.25%, AKD emulsion—0.5%, AKD emulsion—0.75%, AKD emulsion—1.0%, and AKD emulsion—1.25%. All analyses below are based on emulsions in this range of concentrations unless otherwise noted.

### 2.4. Characterization of the AKD Emulsions

#### 2.4.1. Rheological Properties Measurement

The viscoelasticity of the emulsions was recorded using a rheometer (TA ARES-G2, New Castle, DE, USA). The appropriate amounts of AKD emulsions were placed on the plate geometry (25 mm parallel plate diameter, 1 mm gap width) [21]. The shear range was increased from 0.1 to 100 s^−1^ to measure the viscosity. The elastic modulus (G′) and loss modulus (G″) were recorded, and the emulsion frequency sweep was also studied.

#### 2.4.2. Stability Analysis

The stability of the AKD emulsions was evaluated using two approaches. First, the AKD emulsions were stored in 15 mL sample bottles and cooled to room temperature. The static stability of the AKD emulsions was observed and recorded using a digital camera. Second, the simulated long-term physical stability of AKD emulsions was evaluated using a LUM full-featured stability analyzer (LUMisizer, Berlin, Germany). One milliliter of sample was injected into the bottom of the sample plate of the PC tube (110-132 XX). The centrifugal speed was set to 4000 r/min, and the characteristic line of the transmittance was recorded every 5 s for a total of 100 times.

#### 2.4.3. Microstructure of the AKD Emulsions

The morphology of the AKD emulsions was observed using optical microscopy (Leica DM750P, Burladingen, Germany) and a HD camera (Leica MC170, Wetzlar, Germany) with a 20× objective lens. In addition, SEM was used to study the size and morphology of AKD emulsion—1.25% obtained after freeze drying.

#### 2.4.4. Emulsion Droplets Diameter

The diameter of emulsion droplets was measured using a particle size analyzer (Malvern Zetasizer Nano ZS, Worcestershire, UK) using dynamic light scattering.

### 2.5. Preparation of the Packaging Paper and Evaluation of Hydrophobicity

The cellulose paper was prepared on a PTI Rapid-Köthen Blattbildner-Sheet Former (Frank-PTI, Birkenau, Germany) [7,22]. A 200 mL 1 wt% APMP suspension was stirred with a motor stirrer (IKA RW20, Burladingen, Germany) at a speed of 750 r/min, and then the prepared AKD emulsions (0.2 wt% AKD to dry APMP) and retention agent were added gradually. The paper was made from the prepared suspension on the sheet former and dried at 105 °C for 30 min.

The hydrophobicity of the cellulose paper was measured using the liquid permeation method according to China National Standard GB/T 460-2008 [23]. The contact angle of the paper was measured using a contact angle meter (Minsks JC2000D, Xi’an, China) and recorded using a digital CCD camera.

## 3. Results and Discussion

### 3.1. Characterization of the Guar Gel

The guar gel was prepared by cross-linking guar gum molecules in borax aqueous solution at room temperature using a one-step method. Gel formed by guar gum and borax has been prepared by many researchers [16,17,24]. The synthesis mechanism of guar gel is shown in Figure 1a. Sodium borate (Na_2_B_4_O_7_·10H_2_O), dissolved in water, can be hydrolyzed to boric acid and borate ions (B(OH)^4−^), forming a buffer solution of boric acid/borate. Due to the presence of the guar gum cis-hydroxyl groups in the side chain of galactose, a supramolecular guar gel can be easily formed through a borate/didiol complex between borate ions and hydroxyl groups.

Figure 1b shows the FTIR spectra of guar gum (blue line) and guar gel (red line) in a range from 400 to 4000 cm^−1^. Guar gum has a characteristic broad absorption band at 3436 cm^−1^ due to the stretching vibration of O-H, and the band at 2926 cm^−1^ is associated with the stretching vibration of alkane C-H. The peaks at 1648, 1423 and 1152 cm^−1^ are attributed to H-OH bending, C-H bending and C-O-C stretching vibrations, respectively [17,24]. For guar gel, the broad peak at 3436 cm^−1^ sharpened and moved slightly toward longer wavelengths compared to the blue line, possibly because intermolecular hydrogen bonds formed between guar gum and borax during the formation of covalent borate/didiol [16]. The band at 1423 cm^−1^ (C-H bending) broadens and is covered.

The morphology of the freeze-dried guar gel aqueous solution was observed using SEM (Figure 1c,d). Figure 1c displays the three-dimensional network structure in guar gel, which comprises both the membrane and linear cross-linked structures [25]. The membrane structure illustrated in Figure 1d might be the ordered aggregation of guar macromolecules that were cross-linked with borax [16,17].

### 3.2. Effect of Guar Gel on Emulsion Properties

#### 3.2.1. Effect of Guar Gel Concentration on the Rheological Properties of the AKD Emulsions

The rheological properties of emulsions have an important impact on their utilization and long-term stability [26,27]. The effect of guar gel concentration on the rheological properties of the freshly prepared AKD emulsions was investigated by shear-stress controlled rheometry. Figure 2a displays the flow curves of the AKD emulsions. The viscosity curves exhibited pronounced shear-thinning behavior and high viscosities at a low shear rate (typical non-Newtonian pseudoplasticity behavior) [27], which is similar to the pseudoplastic flow characteristics observed in emulsions stabilized by guar gum and whey protein [28]. This behavior is typical of weak associative interactions, suggesting that a weak droplet network structure was formed [27]. As the concentration of guar gel increased, the emulsion viscosity increased, which may be attributed to the increasing amount of guar gel covering the surface of the emulsion droplets. These results are consistent with previous reports on the rheological properties of polysaccharide gel-stabilized emulsions [29,30,31].

The energy storage modulus (G′) and loss modulus (G″) were determined by a frequency sweep measurement. G′ and G″ reflect the energy stored in elastic or solid components and lost in viscous or liquid components, respectively. The crossing point of the modulus indicates that the material has changed from liquid to solid. When G″ is larger than G′ before the crossing point, the material mainly undergoes viscous deformation, which is liquid. When G′ is larger than G″ after the crossing point, the material primarily undergoes elastic deformation, which is solid. When G′ and G″ are equal, the material is in a semisolid form [32]. As shown in Figure 2b–f, when the angular frequency was low, G′ was lower than G″, showing that all the characteristics of the emulsion were similar to those of a liquid. With the increase in angular frequency, G′ was slightly higher than G″, indicating that all the emulsions mainly demonstrated elastic properties and that a weak gel network was formed between the nanoparticles and the oil droplets [33,34]. This effect can be ascribed to the increased flexibility and propensity for entanglement of the guar gel in an aqueous phase, and the structure formed by the molecular chains was not easily destroyed, thereby forming dense networks [31,33,35].

#### 3.2.2. Effect of Guar Gel Concentration on Emulsion Stability and Droplet Diameter

Emulsion stability and droplet diameter are two important parameters related to the application of AKD emulsions [36]. The influence of guar gel concentration on the stability and droplet size of the AKD emulsions is shown in Figure 3.

To observe storage stability, the AKD emulsions were statically stored at room temperature (25 ± 0.5 °C). All the AKD emulsions were homogeneous milky liquids, and only a small amount of water precipitated at the bottom of the AKD emulsions stabilized by the low concentration guar gel after 72 h (Figure 3a). The static stability of the AKD emulsions rose slightly with increasing guar gel concentration, which was attributed to the improvement of the AKD emulsion viscosity and the enhancement of repulsion between the oil droplets.

To investigate long-term physical stability, the AKD emulsions were centrifuged with a stability analyzer at 25 ± 0.5 °C. Centrifugation can accelerate the creaming of the emulsions and remove water and oil phases from the three-dimensional continuous phase network. The transmittance curves changed with the centrifugal cycle time, which is marked with rainbow colors, and the narrower the transmittance curves were, the higher the stability [37]. The fingerprints of AKD emulsion—0.25%, AKD emulsion—0.5%, AKD emulsion—0.75%, AKD emulsion—1.0%, and AKD emulsion—1.25% are shown in Figure 3b–f. The transmittance curves of all AKD emulsions were narrow under stringent test conditions, indicating that all AKD emulsions had good long-term stability. Furthermore, as the guar gel concentration increased, the transmittance curves shifted to the left and tapered gradually, indicating that high stability AKD emulsions with less creaming and more coalescence stability were obtained. The increase in guar gel concentration strengthened the three-dimensional network structure among the emulsion droplets, making the system more stable [38].

The droplet diameter and distribution of AKD emulsions are two important parameters for the application during lignocellulosic paper sizing. It is generally believed that AKD emulsions have a good sizing effect only when more than 90% of the droplets are smaller than 1 μm in diameter [39]. The average diameter and distribution of the AKD emulsion droplets are shown in Figure 4. All the emulsion droplets were spherical in shape, and the average diameter of the droplets decreased from 1223 to 642 nm with increasing guar gel concentration (Figure 4a). The droplets were smaller than those of the previously reported AKD emulsion without surfactant [22,40]. When the guar gel concentration was less than 0.75%, the AKD emulsion droplets tended to flocculate. As the guar gel concentration increased, the AKD emulsion droplets became more uniform and dispersed more evenly. More importantly, all the droplets were smaller than 1 µm, which is very beneficial for the sizing. Only when a sufficient amount of emulsifier is adsorbed on the oil–water interface can an emulsion with small droplets and a high specific surface area be obtained [41]. Furthermore, guar gel, dispersed in the aqueous solution, can lead to steric hindrance among the droplets, which favors improving the aggregate stability of the emulsion [42].

### 3.3. Structure of the AKD Emulsions

A schematic diagram of the AKD emulsions is shown in Figure 5a. The mixture of aqueous guar gel and AKD oil were homogenized under high-speed shearing. During the emulsification process, the guar gel was broken into particles and adsorbed on the oil–water interface. After the external force was eliminated, the guar gel turned into an interface film due to its self-healing property, providing an ideal protective shielding for AKD droplets and improving emulsion stability [43]. The freeze-dried AKD emulsion—1.25% was observed by SEM. From the image in Figure 5c, a few particles smaller than 1 μm can be observed, which is because the large particles can be more clearly observed in the guar film coated on the surface of the particles, whereas the small particles can be easily destroyed by electron microscopy. As shown in Figure 5b,c, the guar gel tightly wrapped around the AKD droplets, forming spherical droplets with a smooth surface, which is beneficial for improving the emulsion stability and reducing droplet coalescence. Notably, there were connections between droplets, allowing the emulsion system to form a three-dimensional network structure. The network structure can improve the viscosity of the emulsion and effectively inhibit droplet coalescence, which is beneficial for improving emulsion stability [22].

The guar gel network, cross-linked via weak hydrogen bonding, is easily broken under mechanical agitation and thus does not hinder the sizing process of AKD emulsions. Moreover, guar gel has high self-healing properties [43], which are conducive to maintaining stability during transportation and storage.

### 3.4. The Hydrophobicity of Paper Sized by AKD Emulsions

Figure 6 displays the wettability and water resistance of the packaging paper prepared under the same sizing conditions with different AKD emulsions. Deionized water (0.05 mL) was dropped onto the prepared paper, and the contact angles photographed at specific time intervals were used to evaluate the wettability and water resistance of the paper sized by the prepared emulsion [44]. Furthermore, the contact angle provided quantitative information on the physical phenomenon of sizing: the larger the initial angle is, the smaller the decrease rate of the contact angle over time, and the better the sizing performance of the slurry [45]. The initial contact angles of all the paper samples were larger than 90°, indicating that the papers possessed a hydrophobic surface, which is due to the large hydrophobic chains existing on the surface of the AKD particles, making larger contact angles easily on the cellulose fibers. At about 100 °C in the drying section of the papermaking process, the AKD droplets melt into a liquid, which then spreads over the fiber’s surface. By virtue of its β-lactone ring, the AKD molecule is able to chemically react with hydroxyl of cellulose, forming a β-keto ester, and the hydrophobic chain in AKD protruded out of the fibers [46,47]. The initial contact angle increased with increasing guar gel concentration, and the initial contact angle of AKD emulsion—1.25% was as high as 129°. With the extension of contact time, the contact angles of the paper samples decreased. In addition, the contact angles of the low guar gel concentration decreased significantly faster than those of the high guar gel concentration, which decreased to only 4° within 900 s for AKD emulsion—1.25%. The result shows that the paper had strong hydrophobicity and the AKD emulsions showed a better sizing performance.

Figure 6b illustrates the liquid permeation sizing degree of the packaging paper, which demonstrated a good liquid penetration inhibition. As the guar gel concentration increased, the sizing degree also increased. The paper sized by AKDE—1.25% attained a sizing degree of 224 s, which was higher than that of the previously reported AKD emulsion at the same application conditions. The results of the contact angle and sizing degree suggest that the higher the guar gel concentration was, the better the water resistance of the packaging paper, which is in line with the AKD emulsion sizing law. It is generally believed that the smaller the emulsion droplets are, the more uniform the droplet distribution, the better the stability, and the greater the AKD emulsion sizing effect [7,40].

## 4. Conclusions

In summary, we successfully obtained AKD emulsions with high sizing efficiency stabilized by guar gel. The guar gel, prepared by hydrogen bond cross-linking sodium tetraborate and guar gum, was used as the emulsifier to prepare the AKD emulsions. By increasing the guar gel concentration, the AKD emulsion stability was improved, the diameter of the average droplet was reduced, and the AKD emulsion viscosity was increased. The AKD emulsion—1.25% proved to be an excellent emulsion with a minimum droplet diameter of 642 nm and a high stability even under high-speed centrifugal conditions.

The guar gel effectively adsorbed on the oil–water interface to form a gel film covering the oil droplets, which formed steric hindrance that prevented emulsion droplet coalescence, resulting in small droplets. Excessive guar gel could form a three-dimensional network, which provided a spatial barrier and architectural support between droplets and enhanced long-term storage stability. Moreover, with the increase in guar gel concentration, the water resistance of the packaging paper improved. This study has provided a simple method to develop solid oil emulsions using gels without surfactants. AKD emulsions are expected to improve sizing efficiency and be used in high water-resistance paper.

## Figures and Tables

**Figure 1 polymers-15-04669-f001:**
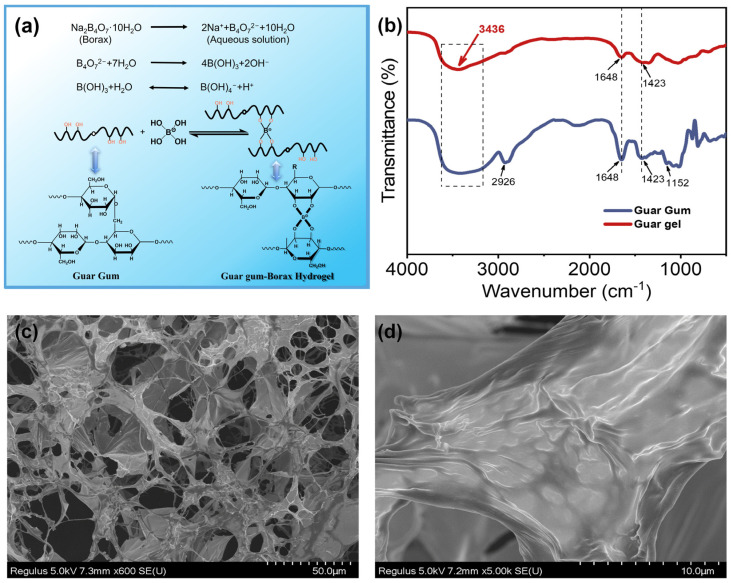
(**a**) Interaction between borax and guar gum; (**b**) FTIR spectra of guar gum and guar gel; (**c**,**d**) SEM images of the freeze-dried guar gel.

**Figure 2 polymers-15-04669-f002:**
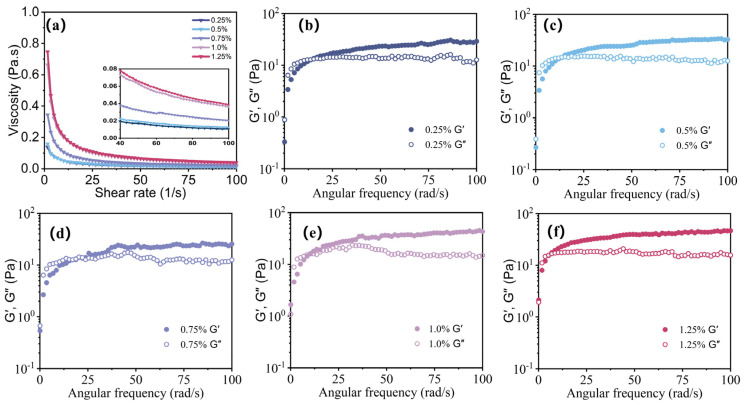
(**a**) Shear rate dependence of viscosity for AKD emulsion—0.25%, AKD emulsion—0.5%, AKD emulsion—0.75%, AKD emulsion—1.0%, and AKD emulsion—1.25%. (**b**–**f**) Dynamic frequency sweep of AKD emulsion—0.25%, AKD emulsion—0.5%, AKD emulsion—0.75%, AKD emulsion—1.0%, and AKD emulsion—1.25%.

**Figure 3 polymers-15-04669-f003:**
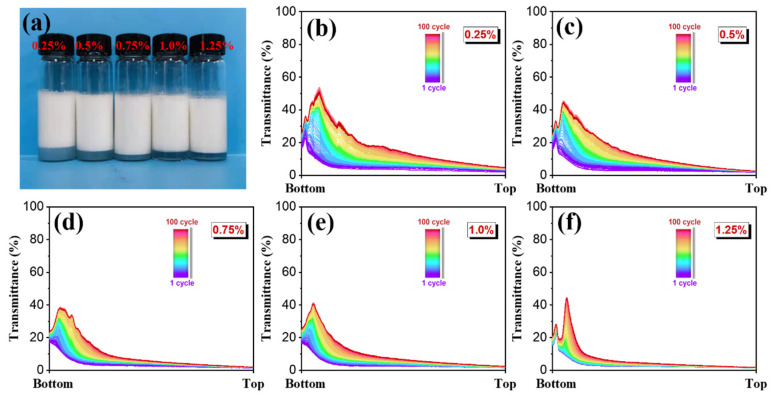
(**a**) Digital photos of AKD emulsions statically stored for 72 h at room temperature; (**b**–**f**) transmittance profiles of AKD emulsion—0.25%, AKD emulsion—0.5%, AKD emulsion—0.75%, AKD emulsion—1.0%, and AKD emulsion—1.25% after centrifugation for 100 cycles at 4000 rpm.

**Figure 4 polymers-15-04669-f004:**
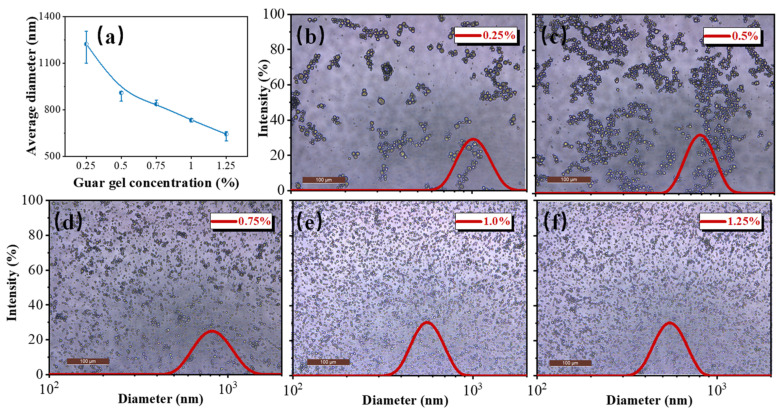
(**a**) Effect of guar gel concentration on the average diameter of AKD emulsions. (**b**–**f**) Digital photos and diameter distribution of AKD emulsion—0.25%, AKD emulsion—0.5%, AKD emulsion—0.75%, AKD emulsion—1.0%, and AKD emulsion—1.25%.

**Figure 5 polymers-15-04669-f005:**
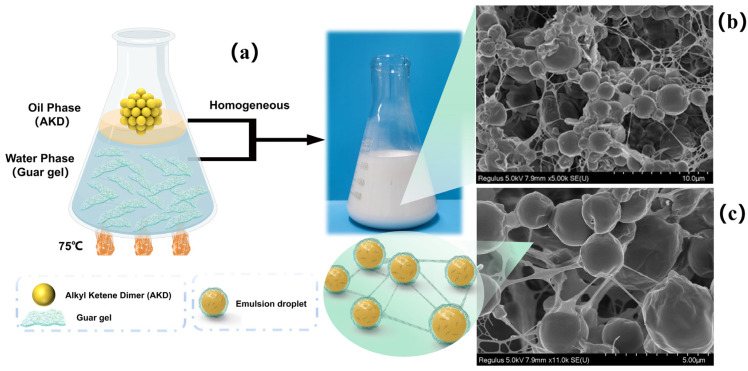
(**a**) Schematic diagram of the AKD emulsions, (**b**,**c**) SEM images of AKD emulsion—1.25%.

**Figure 6 polymers-15-04669-f006:**
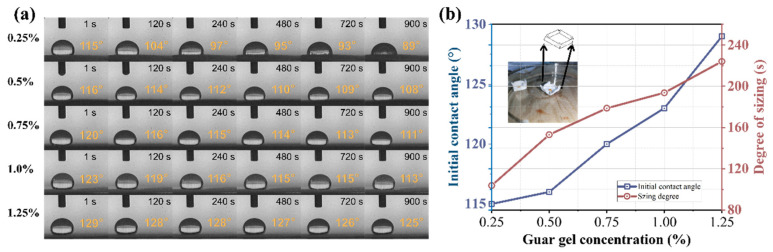
(**a**) Contact angle images of paper sized by AKD emulsions. (**b**) Effect of the guar gel concentration on the sizing degree and initial contact angle of the paper.

## Data Availability

The data presented in this study are available on request from the corresponding author.

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
