# Peer review of "AKD Emulsions Stabilized by Guar Gel: A Highly Efficient Agent to Improve the Hydrophobicity of Cellulose Paper"

_polymers, 2023, doi:10.3390/polym15244669_

Round 1

Reviewer 1 Report

Comments and Suggestions for Authors

This paper reported that guar gel was used as emulsifier to prepare the AKD emulsions.  With increasing guar gel concentration, the stability of the AKD emulsions improved, the droplet diameter decreased, and the hydrophobicity, as well as the water resistance of the sized packaging paper were gradually enhanced. My comments are shown in below,

1.     There are many different kinds of emulsifier used for AKD emulsions preparation. Please illustrate and compare the result of this paper with literatures.

2.     The guar gel was formed only through a borate/didiol complex between borate ions and hydroxyl groups. Why the absorption band at approximately 1648 cm-1 shifted to 1644 cm-1, and the band at approximately 1423 cm-1 (C-H bending) nearly disappeared after crosslinking reaction?

3.     The author stated that “…a three-dimensional network is formed. The membrane and linear cross-linked structure exist simultaneously, which may be ascribed to the irregular chains of guar gum being interconnected and interspersed with each other from SEM microimages.” I think that SEM micro-images are macrostructure of guar gel, it is difficult to conclude the microstructure of polymer chains.

4.     Why is that the G’ of 0.75% sample has lowest G’ among those samples?

5.     The guar gel is a crosslinked polymer with 3D structure. This kind of structure is hardly to as the emulsifier between oil and water phase. Because the particles sizes showed in Fig. 4 (< 1 μm) is magnificently different with the SEM micro-images shown in Fig. 5. (~ 2.5 ~ 3 μm) Can the author draw the detail illustration of emulsification mechanism.

6.     There are large amount of hydroxylic group existed in the surface of AKD particle.  Why the contact angles of all the paper samples were larger than 90°?

Author Response

Response to the reviewer 1

We thank the editor and reviewers for carefully reviewing our manuscript “AKD Emulsions Stabilized by Guar Gel: A Highly Efficient Agent to Improve the Hydrophobicity of Cellulose Paper”. We have revised the manuscript carefully according to the reviewer’s comments. Below is a point-by-point response to the reviewers’ comments.

Reviewer 1:

  1. There are many different kinds of emulsifier used for AKD emulsions preparation. Please illustrate and compare the result of this paper with literatures.

Response: Thanks for your excellent suggestion. The different kinds of emulsifier used for AKD emulsions preparation and comparison with other literatures have been added in the revised manuscript.

  1. The guar gel was formed only through a borate/didiol complex between borate ions and hydroxyl groups. Why the absorption band at approximately 1648 cm-1shifted to 1644 cm-1, and the band at approximately 1423 cm-1 (C-H bending) nearly disappeared after crosslinking reaction?

Response: Thanks for your suggestion. According to the questions raised by the reviewers, we carefully compared the FTIR spectra of guar gum and guar gel, founding that there are no change at the peaks of 1648 cm-1, and the peak at 1423 cm-1 does not disappear, but broadens and is covered. The relevant data and analysis have also been modified in the revised manuscript.

  1. The author stated that “…a three-dimensional network is formed. The membrane and linear cross-linked structure exist simultaneously, which may be ascribed to the irregular chains of guar gum being interconnected and interspersed with each other from SEM microimages.” I think that SEM micro-images are macrostructure of guar gel, it is difficult to conclude the microstructure of polymer chains.

Response: Thanks very much for your suggestion. According to the reviewer's suggestion, we have modified this description in the revised manuscript.

  1. Why is that the G’ of 0.75% sample has lowest G’ among those samples?

Response: Thanks very much for your questions. The G' of all samples were not significantly different at low concentrations, while gradually increase in high concentration.

  1. The guar gel is a crosslinked polymer with 3D structure. This kind of structure is hardly to as the emulsifier between oil and water phase. Because the particles sizes showed in Fig. 4 (< 1 μm) is magnificently different with the SEM micro-images shown in Fig. 5. (~ 2.5 ~ 3 μm) Can the author draw the detail illustration of emulsification mechanism.

Response: Thanks very much for your suggestion. We have investigated various AKD emulsions without surfactants, such as modified calcium carbonate particles, and modified Laponite. Compared to prior studies (Fig. A1) and published literature 1, 2, the AKD emulsion stabilized by guar gum had smaller droplets and excellent stability. As illustrated in Fig5 c, the AKD droplets covered with a film.

The SEM image in Figure 5b has been replaced to more precisely illustrated the droplets size. AKD has a low melting point of 40-60 °C, which makes it easily destroyed by electron beam, especially the small droplets. It is difficult to capture images that include all the tiny droplets using a SEM. This is what causes the variation in droplet sizes between Figure 4 and 5.

We have found that guar gum can be employed to stabilize AKD, however the emulsification process is intricate, so we will strive to explore the stability mechanism in future research.

Figure A1 SEM image of AKD emulsion stabilized by modified calcium carbonate particles.

  1. Zhao, Q., et al., Unique alkyl ketene dimer Pickering-based dispersions: Preparation and application to paper sizing. Journal of Applied Polymer Science, 2018. 135(4): 45730
  2. Chen, X., et al., A stable AKD-in-water emulsion: Stabilized by MSG-modified laponite nanoparticle. Journal of Dispersion Science and Technology, 2017. 38(7): 1067-1072
  3. There are large amount of hydroxylic group existed in the surface of AKD particle.  Why the contact angles of all the paper samples were larger than 90°?

Response: Thanks for your suggestion. The large amount of hydroxylic group existed in the surface of AKD droplets favored the retention of AKD on the cellulose fibers. At about 100°C in the drying section of the papermaking process, the AKD droplets melt into a liquid, which then spreads over the fiber surface. By virtue of its β-lactone ring, the AKD molecule is able to chemically react with hydroxyl of cellulose, forming a β-keto ester, and the hydrophobic chain in AKD protruding out of the fibers (Figure A2). This is the widely accepted AKD sizing mechanism in the papermaking industry.

Figure A2 AKD reaction with cellulose.

Reviewer 2 Report

Comments and Suggestions for Authors

The experimental article "AKD emulsions stabilized by guar gel: a highly efficient agent to improve the hydrophobicity of cellulose paper" fully corresponds to the Polymers publication profile. The material of the article is set out competently, consistently and provided with the necessary number of illustrations. 

In this paper the authors have successfully solved the problem of obtaining ACD emulsion with high bonding efficiency stabilised with guar gel. It is shown that the water resistance of the packaging paper improved with increasing guar gel concentration. The authors conclude that AKD emulsions will improve the bonding efficiency and be used in papers with high water resistance

The authors have sufficiently justified the relevance of the study. And the results of the study look very optimistic. However, I wonder whether the economic calculation of this technological solution was made? How high is the probability of implementation of this technology in industry? If there are such calculations - they would improve the article and broaden the interest of readers to the article.

Author Response

Response to the reviewer 2

We thank the editor and reviewers for carefully reviewing our manuscript “AKD Emulsions Stabilized by Guar Gel: A Highly Efficient Agent to Improve the Hydrophobicity of Cellulose Paper”. We have revised the manuscript carefully according to the reviewers’ comments. Below is a point-by-point response to the reviewer’ s comments.

Reviewer 2:

The experimental article "AKD emulsions stabilized by guar gel: a highly efficient agent to improve the hydrophobicity of cellulose paper" fully corresponds to the Polymers publication profile. The material of the article is set out competently, consistently and provided with the necessary number of illustrations. 

In this paper the authors have successfully solved the problem of obtaining ACD emulsion with high bonding efficiency stabilised with guar gel. It is shown that the water resistance of the packaging paper improved with increasing guar gel concentration. The authors conclude that AKD emulsions will improve the bonding efficiency and be used in papers with high water resistance.

The authors have sufficiently justified the relevance of the study. And the results of the study look very optimistic. However, I wonder whether the economic calculation of this technological solution was made? How high is the probability of implementation of this technology in industry? If there are such calculations - they would improve the article and broaden the interest of readers to the article.

Response: According to the reviewer’s suggestion, the cost of emulsifiers have been calculated as follows: the unit price of guar gum and borax is 40 and 10 ¥/kg, respectively. There are 12.5% AKD per ton of emulsion, which is 125 kg. Taking the sample AKDE-1.25% as an example, there are 125*1.25% kg of guar gum and 2*125*1.25% kg of the bora, and the cost of emulsifier is 93.75 ¥ per ton of emulsion. However, the cost of conventional surfactant and cationic starch emulsifier is 150-180 ¥ per ton of emulsion, and therefore our emulsifiers researched in this work have advantage on cost.

Round 2

Reviewer 1 Report

Comments and Suggestions for Authors

The author has revised the manuscript according to the reviewers' comments. Therefore, I think it can be accepted for publication.